# Hepatitis B virus infection status and associated factors among health care workers in selected hospitals in Kisumu County, Kenya: A cross-sectional study

**Frankline Otieno Mboya**[1]*, **Ibrahim I. Daud**[1], **Raphael Ondondo**[2], **Daniel Onguru**[1]

**1** Jaramogi Oginga Odinga University of Science and Technology, Bondo, Kenya, **2** Masinde Muliro University of Science and Technology, Kakamega, Kenya

\* otifrankline@gmail.com

**Data Availability Statement:** All data collected have been reported in this manuscript.

## Abstract

Poorly managed medical waste produced at the health facilities are potential source of infections including occupational exposure to Hepatitis B Virus (HBV). This study evaluated the prevalence of HBV infection among healthcare workers (HCWs) in Kisumu County. We determined prevalence of HBV infections among 192 HCWs from nine purposively selected high-patient volume public hospitals in Kisumu County. A structured questionnaire was administered, and 4.0 ml of venous blood sample collected for Hepatitis B surface antigen (HBsAg), hepatitis B surface antibody (anti-HBs) and total hepatitis B core antibody (anti-HBc) testing using enzyme immunoassay (EIA). Of 192 HCWs sampled, 52.1% were males and the median participants age was 34.4 years with interquartile range (IQR) of 11 (28–39) years. Most participants (44%) had worked for between 1–5 years. There was low HBV vaccine uptake with 35.9% completing the required 3 doses, while 40.6% had never been vaccinated. HBV prevalence was 18.8% (36/192), prevalence of past resolved infection was 25.5% (49/192), while 37.5% (72/192) of HCW had evidence of vaccine-derived immunity and 17.7% (34/192) were susceptible. HBV prevalence among HCW who had worked for less than one year and those who had never been vaccinated was 37.5% and 35.9% respectively. Significant risk of HBV lifetime exposure was noted among HCWs with one vaccine dose, those with no known exposure, while highest in those with knowledge on HBV transmission (aOR, 7.97; 95% CI, 2.10–153.3, p-value = 0.008). HCWs who had received ≥2 doses of HBV vaccine (aOR, 0.03; 95% CI, 0.01–0.10, p-value = <0.0001) had significant HBV protection. Duration of service was not associated with HBV among HCWs. HBV prevalence was high among HCWs from nine high patient volume public hospitals in Kisumu County. Efforts to strengthen HBV vaccination uptake and dose completion are needed to reduce HBV infections among HCWs.

## Background

Hepatitis B Virus (HBV) infection is a global public health problem with about 2 billion people estimated to have evidence of past or present HBV infection, and 240 million being chronic

**Funding:** The author(s) received no specific funding for this work.

**Competing interests:** The authors have declared that no competing interests exist.

carriers of Hepatitis B surface Antigen (HBsAg), the highest prevalence >5% noted in sub Saharan Africa, East Asia and Amazon Basin of South America [1, 2]. In 2016, mortality due to viral hepatitis was about 1.4 million with 47% of the deaths due to HBV infection [3, 4]. Prevalence of HBV infection in Africa is on average more than 10%, classifying the region as one of high endemic area [5–7]. HBV prevalence is reportedly higher among healthcare workers (HCWs); who include physicians, nurses, clinicians, laboratorians, morticians, medical counsellors, cleaning, and hospital waste management staff [8–11]. Modes of HBV transmission is perinatally, through percutaneous and sexual exposures. In health care settings (HCS) occupational exposures occurs through needle or sharp object injury, mucous membrane and non-intact skin [8, 12]. Hepatitis B vaccination has proven to be a key strategy to prevent infection, the Kenyan Government has a policy where infants are given $1^{st}$ $2^{nd}$ and $3^{rd}$ dose of pentavalent vaccine at 6, 10 and 14 weeks respectively to protect the child from Diphtheria, Pertussis, Tetanus, Hepatitis B and Hib while monovalent hepatitis B vaccine is recommended for most at risk groups like HCWs [13]. There is low hepatitis B vaccine coverage and insufficient immunity to the virus among this high risk HCWs due to uncompleted vaccine dosage coupled with knowledge gaps, inadequate Personal Protective Equipment's (PPEs) and poor waste management process [14–16]. Prevalence of HBV in Kenya is between 3–8%, Human immunodeficiency virus (HIV) disease, blood transfusion and body scarification being potential predictors to HBV infection [17, 18]. HBV is an important occupational hazard to HCWs because of high virulence due to its higher viral load (VL) in the blood, transmissibility in absence of visible blood, longer environmental viability and its availability in several other body fluids [19]. In HCS, concern is high about HBV infections on HCWs due to strong evidence of occupational exposure to sharps and contact with contaminated waste [20, 21]. HBV is considered as Hospital acquired infection (HAI) because unvaccinated HCWs could acquire infection through needle stick injury and contacts, this puts them at higher risk than the general populations [22–24]. The risk of HAI like HBV increases when basic infection prevention and control (IPC) practices in HCS are not well laid out and adhered to. There is limited data on occupational exposure to HBV infection and its prevalence among HCWs in Kenya. This study estimated the prevalence of HBV infection and exposures among HCWs in Kisumu County.

## Methods

### Study aims, design, and setting

We conducted descriptive cross-sectional study between May 2020 and April 2021 to estimate prevalence of HBV infection and its risk factors among HCWs in nine high-patient volume public hospitals in Kisumu County. The selected hospitals provide primary and referral health care services in outpatient (OPD), inpatients (IPD) and special clinics. The hospitals also offer attachment and internship to medical students and have different structural establishments depending on the hospital level. Participants in the study were recruited within the first four months of the study period from outpatient, inpatient, laboratory and mortuary departments within the hospital, waste handlers were from all the service delivery points within the selected hospitals.

### Study population and characteristic of the participants

There were total of 823 HCWs in the selected hospitals, being that these are high-patients volume public hospitals in Kisumu County, they had corresponding high number and carder of HCWs to match the work volume. Sample size representative of the number of HCWs offering services in the selected health facilities was estimated, Cochran formulae for estimating sample size in prevalence studies was used [25] with estimation of 14.6% prevalence, 5% precision at

95% confidence interval. The final sample size was 192 which is 23.3% (192/823) of total health care workforce in the selected facilities. Probability proportional to size (PPS) sampling was used to identify number of HCWs to be included in the study, simple random sampling from duty roster was used to sample the participants from each health facility and service delivery points.

## Data and specimen collection

Sociodemographic data, immunization status, occupational and non-occupational risk factors for hepatitis virus exposure, knowledge on infectious agents and waste management, use and availability of PPEs information was collected from all consenting HCWs using structured questionnaires. The questionnaire was written in English and pretested in Railways dispensary to evaluate the language, comprehension, and flow of questions. From each study participants, qualified and well-trained personnel collected 4.0 ml of venous blood in a 4ml ethylene-diamine-tetra-acetic acid (EDTA) (Becton, Dickinson and Company, Franklin Lakes, New Jersey, USA), the whole blood was triple packed and transported in a cool box within 4 hours of collection to Kenya Medical Research Institute, HIV-Research (KEMRI-HIV-R) Laboratory, Kisumu, Kenya for processing and testing. Whole blood in EDTA was centrifuged at 3,500 rpm for 10 minutes, plasma was harvested and stored at -20°C until testing was done.

## Laboratory testing

Evaluation of HBV infection was based on three biomarkers: hepatitis B surface antigen (HBsAg), antibodies against hepatitis B surface antigen (anti-HBs), and antibodies against total hepatitis B core antigen (anti-HBc). Current HBV infection was determined by testing for HBsAg using Murex HBsAg version 3 kit; immunity to HBV infection was established by testing for anti-HBs using ETI-AB-AUK-3 Diasorin anti-HBs EIA kit; and past exposure to HBV infection was assessed by testing for anti-HBc using Murex anti-HBc (total) kit. All tests were performed by a qualified, trained, and competent assessed laboratory scientists according to the manufacturer's instruction in an ISO 15189 accredited KEMRI HIV-R laboratory. Quality control for both HBsAg, anti HBc and anti HBs tests were performed based on manufacturer's kit instructions and recommendations. HBV current infection was defined as individual's blood is serologically positive for HBsAg while HBV lifetime exposure are individuals whose blood is serologically positive for either HBsAg (current infection) or anti-HBc (may indicate a current or past resolved infection).

## Data analysis

Data was analysed using SPSS version 16.0 (SPSS Inc., Chicago, IL, USA). Results were summarized using descriptive statistics. Logistic regression models were used for bivariate and presented as odd ratios (OR) with 95% confidence intervals (CI). Multivariable analyses were performed for factors attaining p-values ≤0.2 in bivariate analysis to determine independent factors associated with HBV infection (positive for HBsAg or anti-HBc) among HCWs and presented as adjusted OR (aOR). A threshold p-value of less than 0.05 was considered statistically significant. The models were adjusted for age and gender.

## Ethical considerations

The study received ethical approval form JOOTRH ethics review board IERC/JOOTRH/244/20 and National Commission for Science, Technology & Innovation (NACOSTI) granted research permit to conduct the study (licence no: NACOSTI/P/20/6300). Permissions to

collect data from hospitals within Kisumu County was granted by Kisumu County Director of Health. All HBV susceptible HCW were referred for HBV vaccination through Kenya Expanded program for immunization (KEPI).

## Results

### Sociodemographic characteristics and risk factors of the study population

Of the 192 HCWs sampled, 52.1% were males and 78.7% are married, the median participants age was 34.4 years with interquartile range (IQR) of 11 (28–39) years. Most participants (44%) had worked for between 1–5 years. There was low vaccine uptake with 35.9% completing the required 3 doses of HBV vaccine while 40.6% had not been vaccinated, varied results were observed in HCWs IPC training and capacity building: While capacity building on PPEs usage was 73.4%, trainings on waste management and infectious agent found on waste was low at was 30.2% and 32.3% coupled with moderate knowledge on HBsAg transmission, prevention/control, and waste disposal at 56.8%, 53.7% and 30.2% respectively. Whereas 90.6% of HCWs agreed that PPEs are generally available within their work settings, 77.6% felt that they were inadequate. There was low daily usage of PPEs: Apron/dust coat 65.1%, gumboots (waste handlers) 13.5%, gloves 44.8% and mask wearing while on duty 56.3%. There was good access to hand hygiene 96.4% and availability of waste management bins (black, yellow, red) 95.8% with low waste segregation at 29.7% while 67.2% of waste were incinerated at site or networked to facilities with incinerator. Higher proportion (53.1%) of HCWs had either contact or needle stick injury exposure (Table 1).

### Prevalence of HBV biomarkers among HCWs in Kisumu County, 2020 (N = 192)

Among the 192 HCWs tested, the point prevalence of HBV (current infection) was 36 (18.8%) while lifetime prevalence (developed immunity because of natural infection) was 49 (25.5%). There were 72 (37.5%) HCWs who developed immunity after vaccination while 34 (17.7%) were susceptible (never infected and had no evidence of immunization) (Table 2).

Highest prevalence of HBsAg was in HCWs who had worked for less than one year 37.5% (95% CI: 8.5–75.5), HBV unvaccinated HCWs 35.9% (95% CI: 25.3–47.6), HCWs who has had blood transfusion 33.3% (95% CI: 0.84–90.6) and HIV testing counselors 29.4% (95% CI: 10.3–56).

Anti HBc prevalence was highest among HCWs with one dose of HBV vaccine 83.3% (95% CI: 65.3–94.4), those with less than one year in service 75.0% (95% CI: 34.9–96.8), waste handlers not using gumboots 68.8% (95% CI: 50.0–83.9) and HBV unvaccinated HCWs 66.7% (95% CI: 55.1–76.9).

There is moderate HBV immunity or recovery level among HCWs, the carders with highest anti HBs positivity were laboratory scientist 74.6% (95% CI: 61.0–85.3), clinical officers 65.5% (95% CI:45.7–82.1) and Nursing officers 64.1% (95% CI: 47.2–78.8). HTS counselors had the lowest immunity or recovery level at 35.3% (95% CI: 14.2–61.7) (Table 3).

### HBV infection by vaccine uptake

HBV infection was highest among HCWs who had not received any dose of HBV vaccine (35.9%), those who received one dose of vaccine had 13.3% infection rates and those who received two doses had 26.7% of HBV infection. Notably, none of the 69 HCWs who reported receiving all the three required doses of HBV vaccination had HBV infection (Fig 1).

**Table 1. Socio-demographic characteristics and risk factors of the study population (N = 192).**

| Variables | Frequency | Percent (%) |
|---|---|---|
| **Sex** | | |
| Female | 92 | 47.92 |
| Male | 100 | 52.08 |
| **Age (years)** | | |
| 20–29 | 65 | 33.85 |
| 30–39 | 83 | 43.23 |
| 40–49 | 23 | 11.98 |
| ≥50 | 21 | 10.94 |
| **Marital status** | | |
| Single | 37 | 19.27 |
| Married | 151 | 78.65 |
| Widowed/Separated/Divorced | 4 | 2.08 |
| **Years of service (years)** | | |
| Less than 1 | 8 | 4.17 |
| 1–5 | 85 | 44.27 |
| 6–10 | 51 | 26.56 |
| >10 | 48 | 25 |
| **Cadre** | | |
| Doctors | 8 | 4.17 |
| Clinical Officer | 29 | 15.1 |
| Nursing officer | 39 | 20.31 |
| Laboratory technologist | 55 | 28.65 |
| Mortuary attendance | 7 | 3.65 |
| HIV testing counsellor | 17 | 8.85 |
| Medical waste handlers | 37 | 19.27 |
| **HBV Vaccine uptake** | | |
| 3 doses | 69 | 35.9 |
| 2 doses | 15 | 7.8 |
| 1 dose | 30 | 15.6 |
| Not vaccinated | 78 | 40.6 |
| **History of exposure** | | |
| Needle Stick Injury | 91 | 47.4 |
| Contact Exposure | 11 | 5.7 |
| No Known Exposure | 90 | 46.9 |
| **History of Blood transfusion** | | |
| No | 189 | 98.4 |
| Yes | 3 | 1.6 |
| **Knowledge on HBsAg Transmission** | | |
| No | 83 | 43.2 |
| Yes | 109 | 56.8 |
| **Knowledge on HBsAg Prevention and control** | | |
| No | 89 | 46.4 |
| Yes | 103 | 53.7 |
| **Training on PPE full module** | | |
| No | 141 | 26.6 |
| Yes | 51 | 73.4 |
| **Training on infectious agent on waste full module** | | |

*(Continued)*

**Table 1.** (Continued)

| Variables | | Frequency | Percent (%) |
|---|---|---|---|
| | No | 130 | 67.7 |
| | Yes | 62 | 32.3 |
| **Training on waste management full module** | | | |
| | No | 134 | 69.8 |
| | Yes | 58 | 30.2 |
| **Knowledge on waste disposal** | | | |
| | No | 90 | 46.9 |
| | Yes | 102 | 53.1 |
| **Availability of PPEs** | | | |
| | No | 18 | 9.4 |
| | Yes | 174 | 90.6 |
| **Adequacy of PPEs** | | | |
| | No | 149 | 77.6 |
| | Yes | 43 | 22.4 |
| **Daily use of apron/Dust coat** | | | |
| | No | 67 | 34.9 |
| | Yes | 125 | 65.1 |
| **Daily use of gumboots (Waste handlers only)** | | | |
| | No | 32 | 86.5 |
| | Yes | 5 | 13.5 |
| **Daily use of gloves** | | | |
| | No | 106 | 55.2 |
| | Yes | 86 | 44.8 |
| **Daily use of mask when on duty** | | | |
| | No | 84 | 43.8 |
| | Yes | 108 | 56.3 |
| **Access to hand hygiene** | | | |
| | No | 7 | 3.7 |
| | Yes | 185 | 96.4 |
| **Availability of waste management materials (All 3 waste bins)** | | | |
| | No | 8 | 4.2 |
| | Yes | 184 | 95.8 |
| **Proper waste segregation** | | | |
| | No | 135 | 70.3 |
| | Yes | 57 | 29.7 |
| **Waste disposal Method (Incineration)** | | | |
| | No | 63 | 32.8 |
| | Yes | 129 | 67.2 |

Values are presented as numbers and proportions (%).

## Factors associated with current HBV infection

Table 4 shows that female HCWs were more likely to have current HBV infection compared to their male counterparts (aOR, 3.22; 95% CI, 1.06–9.75, p-value< 0.05). Additionally, HCWs without a history of known exposure had increased odds of current HBV infection compared to those with a previous needle stick injury (aOR, 5.37; 95% CI, 1.81–15.92, p-value< 0.001). However, HCWs who reported receiving ≥2 doses of HBV vaccination had reduced likelihood

**Table 2. Interpretation of serologic markers: Hepatitis B virus infection status and corresponding percentages among HCWs in Kisumu country, 2020 (N = 192).**

| Clinical status | HBsAg | Total anti HBs | Total Anti HBc | Action taken | n (%) |
|---|---|---|---|---|---|
| Current infection | Positive | Negative | Positive | linked to hepatitis B directed care | 36 (18.8) |
| Resolved infection (immune after infection) | Negative | Positive | Positive | offered counselling and reassurance | 49 (25.5) |
| Immune (immunization) | Negative | Positive | Negative | offered counselling and reassurance | 72 (37.5) |
| Susceptible (Never infected and no evidence of immunization) | Negative | Negative | Negative | vaccinated through KEPI | 34 (17.7) |
| Isolated core antibody (Indeterminate) | Negative | Negative | Positive | Counselled and referred for further clinical advice | 1(0.5) |

HBsAg, hepatitis B surface antigen; Anti-HBc, hepatitis B core antibody; Anti-HBs, hepatitis B surface antibody.

of current HBV infection. (aOR, 0.05; 95% CI, 0.01–0.20, p-value <0.001) respectively. None of the other sociodemographic characteristics were associated with current infection of HBV among HCWs.

## Factors associated with lifetime exposure to HBV infection among HCW in Kisumu County, 2020

In logistic regression analysis (Table 5), HCWs who received a single dose of HBV vaccination had increased likelihood of lifetime exposure to HBV infection compared to HCWs without history of vaccination (aOR, 6.25; 95% CI, 1.29–30.30, p-value<0.05). Conversely, HCWs who reported receiving ≥2 doses of HBV vaccination had reduced likelihood of lifetime exposure to HBV infection compared to those without HBV vaccination (aOR, 0.03; 95% CI, 0.01–0.10, p-value = <0.0001) HCWs who reported having knowledge on HBsAg transmission had higher odds of lifetime exposure to HBV infection compared to their counterparts without knowledge on HBV transmission (aOR, 7.97; 95% CI, 2.10–153.39, p-value<0.01). None of the other sociodemographic characteristics were significantly associated with current infection of HBV in HCWs.

## Discussions

Indeed the burden of HAI like Hepatitis B infection is high in developing sub-Saharan African countries like Kenya [26]. Despite the availability of guidelines and treatment options, occupational risks related to hepatitis virus exposure is still a major concern for those who handle hospital waste like HCWs [27]. This study documented high prevalence of current and resolved HBV infection among HCWs in Kisumu County, Kenya at 18.8% and 25.5% respectively. The subpopulations with the highest HBV infection are: HCWs who had worked for less than 1 year at 37.5%, HBV unvaccinated HCWs at 35.9%, HCWs with previous history of blood transfusion at 33.3% and HIV testing counselor at 29.4%. The HBV prevalence among HCWs in this study is higher compared to 2.7% prevalence in general population [28]. This prevalence is higher than what was found in other studies done in Kenya and Africa: pooled prevalence in Africa 6.8%, Kenya 4.0%, southern Ethiopia 1.3%, north west Ethiopia 6.0%, Tripoli Libya 2.3% and 6.3% in Addis Ababa [5, 6, 7, 14, 27, 29]. A study in Kisumu, Siaya and Homa-bay countys in Kenya found that the prevalence of HBV among adolescent was 3.4% while dual infection of HBV and HIV among patients presenting with jaundice in Kisumu county hospital clinic, one of our study site was 47% [30, 31]. The high prevalence in this study could be attributed to increased HBV risk of exposure on HCWs as highlighted in this study where 53.7% of HCWs has had either needle stick or contact exposure. Other factors noted in this study which could contribute the high prevalence seen are: inadequate trainings, knowledge

**Table 3. Hepatitis B virus prevalence by sociodemographic and risk factors among HCW in Kisumu County, 2020 (N = 192).**

| Characteristic | Prevalence of HBsAg Biomarker | | Prevalence of Anti-HBs Biomarker | | Prevalence of Anti-HBc Biomarker | |
|---|---|---|---|---|---|---|
| | N (%) | % (95%CI) | N (%) | % (95%CI) | N (%) | % (95%CI) |
| **Cadre** | | | | | | |
| Doctors | 2/8 | 25.0 (3.19–65.1) | 5/8 | 62.5 (24.5–91.5) | 3/8 | 37.5 (8.5–75.5) |
| Clinical Officer | 6/29 | 20.7 (8–39.7) | 19/29 | 65.5 (45.7–82.1) | 14/29 | 48.3 (29.5–67.5) |
| Nursing officer | 8/39 | 20.5 (9.3–36.5) | 25/39 | 64.1 (47.2–78.8) | 15/39 | 38.5 (23.4–55.4) |
| Laboratory technologist | 7/55 | 12.7 (5.3–24.5) | 41/55 | 74.6 (61.0–85.3) | 18/55 | 32.7 (20.7–46.7) |
| Mortuary attendance | 0/7 | 0.0 (0–41.0) | 4/7 | 57.1 (18.4–90.1) | 4/7 | 57.1 (18.4–90.1) |
| HTS counsellor | 5/17 | 29.4 (10.3–56) | 6/17 | 35.3 (14.2–61.7) | 8/17 | 47.1 (23.0–72.2) |
| Waste handlers | 8/37 | 21.6 (9.8–38.2) | 21/37 | 56.8 (39.5–72.9) | 24/37 | 64.9 (47.5–79.8) |
| **Sex** | | | | | | |
| Male | 12/100 | 12 (6.4–20.0) | 69/100 | 69.0 (59.0–77.9) | 46/100 | 46.0 (36.0–56.3) |
| Female | 24/92 | 26.1 (17.5–36.3) | 52/92 | 56.5 (45.8–66.8) | 40/92 | 43.5 (33.2–54.2) |
| **Age (Years)** | | | | | | |
| 20–29 | 10/65 | 15.4 (7.6–26.5) | 41/65 | 63.1 (50.2–74.7) | 25/65 | 38.5 (26.7–51.4) |
| 30–39 | 15/83 | 18.1 (10.5–28.1) | 56/83 | 67.5 (56.3–77.4) | 35/83 | 42.2 (31.4–53.5) |
| 40–49 | 6/23 | 26.1 (10.2–48.4) | 11/23 | 47.8 (26.8–69.4) | 14/23 | 60.9 (38.5–80.3) |
| ≥50 | 5/21 | 23.8 (8.2–47.2) | 13/21 | 61.9 (38.4–85.1) | 12/21 | 57.1 (34.0–78.2) |
| **Marital status** | | | | | | |
| Single | 7/37 | 18.9 (8.0–35.2) | 20/37 | 54.1 (36.9–70.5) | 15/37 | 40.5 (24.8–57.9) |
| Married | 28/151 | 18.5 (12.7–25.7) | 98/151 | 64.9 (56.7–72.5) | 69/151 | 45.7 (37.6–54.0) |
| Widowed/Separated/Divorced | 1/4 | 25 (0.6–80.6) | 3/4 | 75.0 (19.4–99.4) | 2/4 | 50.0 (6.8–93.2) |
| **Years of service (Years)** | | | | | | |
| <1 | 3/8 | 37.5 (8.5–75.5) | 4/8 | 50.0 (15.7–84.3) | 6/8 | 75.0 (34.9–96.8) |
| 1–5 | 13/85 | 15.3 (8.4–24.7) | 51/85 | 60.0 (48.8–70.5) | 31/85 | 36.5 (26.3–47.6) |
| 6–10 | 10/51 | 19.6 (9.8–33.1) | 38/51 | 74.5 (60.4–85.7) | 26/51 | 51.0 (36.6–65.3) |
| >10 | 10/48 | 20.8 (10.5–35.0) | 28/48 | 58.3 (43.2–72.4) | 23/48 | 47.9 (33.3–62.8) |
| **HBV Vaccine uptake** | | | | | | |
| 1 dose | 4/30 | 13.3 (3.8–30.7) | 20/30 | 66.7 (47.2–82.7) | 25/30 | 83.3 (65.3–94.4) |
| 2 doses | 4/15 | 26.7 (7.8–55.1) | 8/15 | 53.3 (26.6–78.7) | 9/15 | 60.0 (32.3–83.7) |
| 3 doses | 0/69 | 0.0 (0.0–5.2) | 69/69 | 100.0 (94.8–100) | 0/69 | 0.0 (0.0–5.2) |
| Not vaccinated | 28/78 | 35.9 (25.3–47.6) | 24/78 | 30.8 (20.8–42.2) | 52/78 | 66.7 (55.1–76.9) |
| **History of exposure** | | | | | | |
| Needle Stick Injury | 8/91 | 8.8 (3.9–16.6) | 65/91 | 71.4 (61.0–80.4) | 37/91 | 40.7 (30.5–51.5) |
| Contact Exposure | 0/11 | 0.0 (0.0–28.5) | 8/11 | 72.7 (39.0–94.0) | 2/11 | 18.2 (2.3–51.8) |
| No Known Exposure | 28/90 | 31.1 (21.8–41.7) | 48/90 | 53.3 (42.5–63.9) | 47/90 | 52.2 (41.4–62.9) |
| **Blood transfusion** | | | | | | |
| No | 35/189 | 18.5 (13.3–24.8) | 119/189 | 63.0 (55.7–69.9) | 85/189 | 45.0 (37.8–52.4) |
| Yes | 1/3 | 33.3 (0.84–90.6) | 2/3 | 66.7 (9.4–99.2) | 1/3 | 33.3 (0.8–90.6) |
| **Knowledge on HBsAg Transmission** | | | | | | |
| No | 18/83 | 21.7 (13.4–32.1) | 40/83 | 48.2 (37.1–59.4) | 43/83 | 51.8 (40.6–62.9) |
| Yes | 18/109 | 16.5 (10.1–24.8) | 81/109 | 74.3 (65.1–82.2) | 43/109 | 39.5 (30.2–49.3) |
| **Knowledge on HBsAg Prevention and control** | | | | | | |
| No | 20/89 | 22.5 (14.3–32.6) | 45/89 | 50.6 (39.8–61.3) | 46/89 | 51.7 (40.8–62.4) |
| Yes | 16/103 | 15.5 (9.2–24.0) | 76/103 | 73.8 (64.2–82.0) | 40/103 | 38.8 (29.4–48.9) |
| **Training on PPE full module** | | | | | | |
| No | 28/141 | 19.9 (13.6–27.4) | 81/141 | 57.5 (48.9–65.7) | 65/141 | 46.1 (37.7–54.7) |

*(Continued)*

**Table 3.** (Continued)

| | | Prevalence of HBsAg Biomarker | | Prevalence of Anti-HBs Biomarker | | Prevalence of Anti-HBc Biomarker | |
|---|---|---|---|---|---|---|---|
| | Yes | 8/51 | 15.7 (7.0–28.6) | 40/51 | 78.4 (64.7–88.7) | 21/51 | 41.2 (27.6–55.8) |
| **Training on infectious agent on waste full module** | | | | | | | |
| | No | 28/130 | 21.5 (14.8–29.6) | 75/130 | 57.7 (48.7–66.3) | 64/130 | 49.2 (40.4–58.1) |
| | Yes | 8/62 | 12.9 (5.7–23.9) | 46/62 | 74.2 (61.5–84.5) | 22/62 | 35.5 (23.7–48.7) |
| **Training on waste management full module** | | | | | | | |
| | No | 26/134 | 19.4 (13.1–27.1) | 80/134 | 59.7 (50.9–68.1) | 65/134 | 48.5 (39.8–57.3) |
| | Yes | 10/58 | 17.2 (8.6–29.4) | 41/58 | 70.7 (57.3–81.9) | 21/58 | 36.2 (24.0–49.9) |
| **Knowledge on waste disposal** | | | | | | | |
| | No | 21/90 | 23.3 (15.1–33.4) | 49/90 | 54.4 (43.6–65.0) | 47/90 | 52.2 (41.4–62.9) |
| | Yes | 15/102 | 14.7 (8.5–23.1) | 72/102 | 70.6 (60.8–79.2) | 39/102 | 38.2 (28.8–48.4) |
| **Availability of PPEs** | | | | | | | |
| | No | 4/18 | 22.2 (6.4–47.6) | 8/18 | 44.4 (21.5–69.2) | 7/18 | 38.9 (17.3–64.3) |
| | Yes | 32/174 | 18.4 (12.9–25.0) | 113/174 | 64.9 (57.4–72.0) | 79/174 | 45.4 (37.9–53.1) |
| **Adequacy of PPEs** | | | | | | | |
| | No | 31/149 | 20.8 (14.6–28.2) | 89/149 | 59.7 (51.4–67.7) | 74/149 | 49.6 (41.4–58.0) |
| | Yes | 5/43 | 11.6 (3.9–25.1) | 32/43 | 74.4 (58.8–86.5) | 12/43 | 27.9 (15.3–43.7) |
| **Daily use of apron/Dust coat** | | | | | | | |
| | No | 16/67 | 23.9 (14.3–35.9) | 38/67 | 56.7 (44.0–68.8) | 37/67 | 55.2 (42.6–67.4) |
| | Yes | 20/125 | 16.0 (10.1–23.6) | 83/125 | 66.4 (57.4–74.6) | 49/125 | 39.2 (30.6–48.3) |
| **Daily use of gumboots (Waste handlers only)** | | | | | | | |
| | No | 8/32 | 25.0 (11.5–43.4) | 17/32 | 53.1 (34.7–70.9) | 22/32 | 68.8 (50.0–83.9) |
| | Yes | 0/5 | 0.0 (0.0–52.2) | 4/5 | 80.0 (28.4–99.5) | 2/5 | 8.3 (5.3–85.3) |
| **Daily use of gloves** | | | | | | | |
| | No | 18/106 | 17.0 (10.4–25.5) | 74/106 | 69.8 (60.1–78.4) | 56/106 | 52.8 (42.9–62.6) |
| | Yes | 18/86 | 20.9 (12.9–31.2) | 47/86 | 54.7 (43.6–65.4) | 30/86 | 34.9 (24.9–45.9) |
| **Daily use of mask when on duty** | | | | | | | |
| | No | 15/84 | 17.9 (10.4–27.7) | 55/84 | 65.5 (54.3–75.5) | 40/84 | 47.6 (36.6–58.8) |
| | Yes | 21/108 | 19.4 (12.5–28.2) | 66/108 | 61.1 (51.3–70.3) | 46/108 | 42.6 (33.1–52.5) |
| **Access to hand hygiene** | | | | | | | |
| | No | 2/7 | 28.6 (3.7–71.0) | 3/7 | 42.9 (9.9–81.6) | 3/7 | 42.9 (9.9–81.6) |
| | Yes | 34/185 | 18.4 (13.1–24.7) | 118/185 | 63.8 (56.4–70.7) | 83/185 | 44.9 (37.6–52.3) |
| **Availability of waste management materials (All 3 waste bins)** | | | | | | | |
| | No | 2/8 | 25.0 (3.2–65.1) | 4/8 | 50.0 (15.7–84.3) | 4/8 | 50.0 (15.7–84.3) |
| | Yes | 34/184 | 18.5 (13.2–24.9) | 117/184 | 63.6 (56.2–70.5) | 82/184 | 44.6 (37.3–52.1) |
| **Proper waste segregation** | | | | | | | |
| | No | 28/135 | 20.7 (14.3–28.6) | 79/135 | 58.5 (49.7–66.9) | 66/135 | 48.9 (40.2–57.6) |
| | Yes | 8/57 | 14.0 (6.3–25.8) | 42/57 | 73.7 (60.3–84.5) | 20/57 | 35.1 (22.9–48.9) |
| **Waste disposal Method (Incineration)** | | | | | | | |
| | No | 10/63 | 15.9 (7.9–27.3) | 42/63 | 66.7 (53.7–78.1) | 28/63 | 44.4 (31.9–57.5) |
| | Yes | 26/129 | 20.2 (13.6–28.1) | 79/129 | 61.2 (52.3–69.7) | 58/129 | 45.0 (36.2–54.0) |

Values are presented as number (%); CI, confidence interval; HBsAg, hepatitis B surface antigen; Anti-HBc; hepatitis B core antibody; Anti-HCV, antibody hepatitis C virus.

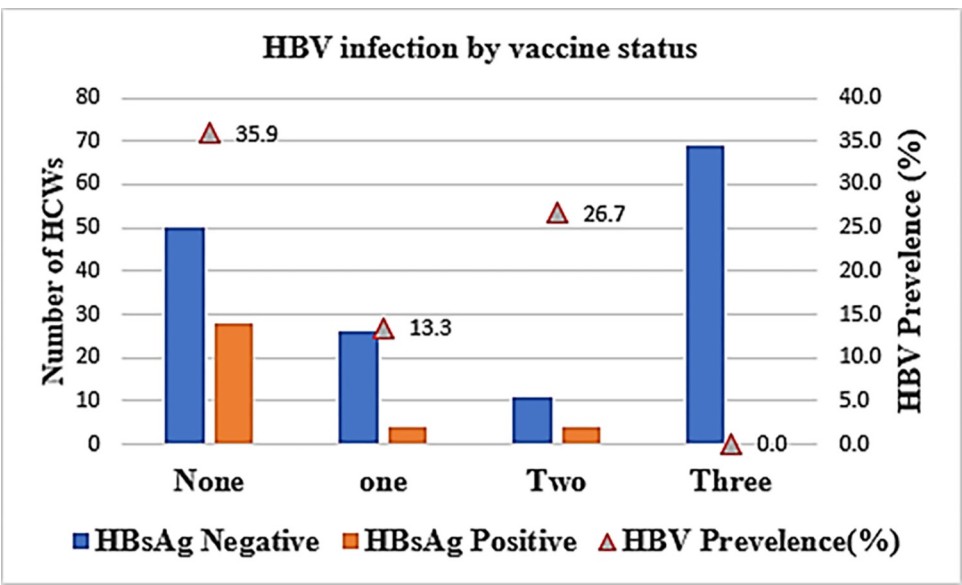

**Fig 1. HBV prevalence by vaccine status among HCWs in Kisumu County.**

gaps, poor infection prevention infrastructure, low Hepatitis B virus vaccine coverage and low-adherence universal IPC measures like inconsistent use of PPEs, improper waste segregations and disposal as shown in (Table 1).

Low training coverage on waste management, infectious agent on waste and inadequacy of PPEs at 30.2%,32.2%, 22.4% respectively could be the cause to high needle stick injury, contact exposure and poor adherence to standard precautions seen in this study. Improperly managed waste poses a health risk to the patients, staff, public and the environments [32]. The findings on inadequate HCWs capacity building, poor adherence on IPC standard and additional precautions are in concurrence with other studies which pointed to poor waste management, lack of training and capacity building to staff [29, 33, 34]. Health care system administrators should ensure adequate PPEs are available, accessible, and properly used by all HCWs, IPC trainings should be done to all health care workers with provision of annual refreshers.

This study found that 37.5% of HCWs have been successfully immunized (recovered from infection or successfully vaccinated) while 17.7% are susceptible (never infected and no evidence of immunization). The carders with high immunization rates were laboratory technologists 74.6%, clinical officers 65.5%, and Nursing officers 64.1%. Laboratory technologists were the highest immunized carder, this concurs with a study in Mozambique where 77% of laboratory technicians were vaccinated [35], the high vaccination rate in this carder may be attributed to the implementation of laboratory quality management system and international organization for standardization ISO 15189 which required that all laboratory personnel to be vaccinated against blood borne pathogens [36]. The low HBV immunity rate and non-completion of vaccine doses is comparable to prior studies showing low HBV vaccine uptake and non-completion of vaccine dosses [5, 11, 29]. The benefit vaccination as HBV infection prevention measure has been documented [37], this study also found HCWs who are fully vaccinated had 0% of infection while higher rates of infection was in unvaccinated HCWs at 35.9%. Findings in this study highlights the need to capacity build HCWs on the benefits of completing the dosage and need to avail the vaccine for all health care workforce. Significant risk of lifetime exposure to HBV infection was noted among HCWs with one vaccine dose, those with no known exposure and highest in those who had Knowledge on HBsAg transmission

**Table 4. Factors associated with current HBV infection among HCWs in Kisumu County.**

| Characteristic | HWC (%) | N (%) | OR (95% CI) | p-value | aOR (95% CI) | p-value |
|---|---|---|---|---|---|---|
| Overall | N (%) | 36 (18.8) | | | | |
| **Cadre** | | | | | | |
| Doctors | 8 (4.2) | 2 (25.0) | 1 | | | |
| Clinical Officer | 29 (15.1) | 6 (20.7) | 0.78 (0.12–4.90) | 0.793 | | |
| Nursing officer | 39 (20.3) | 8 (20.5) | 0.77 (0.13–4.59) | 0.778 | | |
| Laboratory technologist | 55 (28.7) | 7 (12.7) | 0.44 (0.07–2.61) | 0.364 | | |
| Mortuary attendance | 7 (3.7) | 0 (0.0) | | | | |
| HTS counsellor | 17 (9.9) | 5 (29.4) | 1.25 (0.19–8.44) | 0.819 | | |
| Waste handlers | 37 (19.3) | 8 (21.6) | 0.83 (0.14–4.91) | 0.835 | | |
| **Sex** | | | | | | |
| Male | 92 (47.9) | 12 (12.0) | 1 | | 1 | |
| Female | 100 (52.1) | 24 (26.1) | 2.59 (1.21–5.54) | 0.014 | 3.22 (1.06–9.75) | 0.039 |
| **Age (years)** | | | | | | |
| 20–29 | 65 (33.9) | 10 (15.4) | 1 | | | |
| 30–39 | 83 (43.2) | 15 (18.1) | 1.21 (0.51–2.91) | 0.665 | | |
| 40–49 | 23 (12.0) | 6 (26.1) | 1.94 (0.62–6.12) | 0.258 | | |
| ≥50 | 21 (10.9) | 5 (23.8) | 1.72 (0.51–5.76) | 0.38 | | |
| **Marital status** | | | | | | |
| Single | 37 (19.3) | 7 (18.9) | 1 | | | |
| Married | 151 (78.7) | 28 (18.5) | 0.98 (0.39–2.45) | 0.958 | | |
| Widowed/Separated/Divorced | 4 (2.1) | 1 (25.0) | 1.42 (0.13–15.87) | 0.772 | | |
| **Years of service(years)** | | | | | | |
| <1 | 8 (4.2) | 3 (37.5) | 1 | | 1 | |
| 1–5 | 85 (44.3) | 13 (15.3) | 0.30 (0.06–1.42) | 0.128 | 0.56 (0.08–3.98) | 0.56 |
| 6–10 | 51 (26.6) | 10 (19.6) | 0.41 (0.08–1.99) | 0.267 | 1.89 (0.17–21.55) | 0.607 |
| >10 | 48 (25.0) | 10 (20.8) | 0.44 (0.09–2.15) | 0.31 | 1.61 (0.1–25.91) | 0.738 |
| **HBV Vaccine uptake** | | | | | | |
| Not vaccinated | 78 (40.6) | 28 (35.9) | 1 | | 1 | |
| 1 dose | 30 (15.6) | 4 (13.3) | 0.27 | 0.028 | 0.31 (0.06–1.54) | 0.153 |
| ≥2 doses | 84 (43.8) | 4 (4.8) | 0.09 | <0.0001 | 0.05 (0.01–0.20) | <0.0001 |
| **History of exposure** | | | | | | |
| Needle Stick Injury | 91 (47.4) | 8 (8.8) | 1 | | 1 | |
| Contact Exposure | 11 (5.7) | 0 (0.0) | 1 (-) | | 1 (-) | |
| No Known Exposure | 90 (46.9) | 28 (31.1) | 4.69 (2.00–10.98) | <0.001 | 5.37 (1.81–15.92) | <0.001 |
| **Blood transfusion** | | | | | | |
| No | 189 (98.4) | 35 (18.5) | 1 | | | |
| Yes | 3 (1.6) | 1 (33.3) | 2.20 (0.19–24.95) | 0.525 | | |
| **Knowledge on HBsAg Transmission** | | | | | | |
| No | 83 (43.2) | 18 (21.7) | 1 | | | |
| Yes | 109 (56.8) | 18 (16.5) | 0.71 (0.35–1.48) | 0.364 | | |
| **Knowledge on HBsAg Prevention and control** | | | | | | |
| No | 89 (46.4) | 20 (22.5) | 1 | | | |
| Yes | 103 (53.7) | 16 (15.5) | 0.63 (0.31–1.32) | 0.222 | | |
| **Training on PPE full module** | | | | | | |
| No | 141 (26.6) | 28 (19.9) | 1 | | | |
| Yes | 51 (73.4) | 8 (15.7) | 0.75 (0.32–1.78) | 0.514 | | |
| **Training on infectious agent on waste full module** | | | | | | |

*(Continued)*

**Table 4.** (Continued)

| Characteristic | | HWC (%) | N (%) | OR (95% CI) | p-value | aOR (95% CI) | p-value |
|---|---|---|---|---|---|---|---|
| | No | 130 (67.7) | 28 (21.5) | 1 | | 1 | |
| | Yes | 62 (32.3) | 8 (12.9) | 0.54 (0.23–1.27) | 0.156 | 1.53 (0.31–7.57) | 0.604 |
| **Training on waste management full module** | | | | | | | |
| | No | 134 (69.8) | 26 (19.4) | 1 | | | |
| | Yes | 58 (30.2) | 10 (17.2) | 0.87 (0.39–1.93) | 0.725 | | |
| **Knowledge on waste disposal** | | | | | | | |
| | No | 90 (46.9) | 21 (23.3) | 1 | | 1 | |
| | Yes | 102 (53.1) | 15 (14.7) | 0.57 (0.27–1.18) | 0.129 | 0.49 (0.13–1.86) | 0.297 |
| **Availability of PPEs** | | | | | | | |
| | No | 18 (9.4) | 4 (22.2) | 1 | | | |
| | Yes | 174 (90.6) | 32 (18.4) | 0.79 (0.24–2.56) | 0.692 | | |
| **Adequacy of PPEs** | | | | | | | |
| | No | 149 (77.6) | 27 (19.6) | 1 | | 1 | |
| | Yes | 43 (22.4) | 5(11.6) | 0.50 (0.18–1.38) | 0.181 | 0.47 (0.13–1.72) | 0.256 |
| **Daily use of apron/Dust coat** | | | | | | | |
| | No | 67 (34.9) | 16 (23.9) | 1 | | 1 | |
| | Yes | 125 (65.1) | 20(16.0) | 0.61 (0.29–1.27) | 0.185 | 0.57 (0.20–1.64) | 0.297 |
| **Daily use of gumboots (Waste handlers only)** | | | | | | | |
| | No | 32 (86.5) | 5 (13.5) | 1 | | | |
| | Yes | 5 (13.5) | 32 (86.5) | 1(-) | '- | | |
| **Daily use of gloves** | | | | | | | |
| | No | 106 (55.2) | 18 (17.0) | 1 | | | |
| | Yes | 86 (44.8) | 18 (20.9) | 1.29 (0.63–2.67) | 0.486 | | |
| **Daily use of mask when on duty** | | | | | | | |
| | No | 84 (43.8) | 15 (17.9) | 1 | | | |
| | Yes | 108 (56.3) | 21 (19.4) | 1.11 (0.53–2.31) | 0.78 | | |
| **Access to hand hygiene** | | | | | | | |
| | No | 7 (3.7) | 2 (28.6) | 1 | | | |
| | Yes | 185 (96.4) | 34 (18.4) | 0.56 (0.10–3.02) | 0.503 | | |
| **Availability of waste management materials (All 3 waste bins)** | | | | | | | |
| | No | 8 (4.2) | 2 (25.0) | 1 | | | |
| | Yes | 184 (95.8) | 34 (18.5) | 0.68 (0.13–3.52) | 0.645 | | |
| **Proper waste segregation** | | | | | | | |
| | No | 135 (70.3) | 28 (20.7) | 1 | | | |
| | Yes | 57 (29.7) | 8 (14.0) | 0.62 (0.27–1.47) | 0.28 | | |
| **Waste disposal Method (Incineration)** | | | | | | | |
| | No | 63 (32.8) | 10 (15.9) | 1 | | | |
| | Yes | 129 (67.2) | 26 (20.2) | 1.34 (0.60–2.98) | 0.476 | | |

Values are presented as number (%); OR, odds ratio; aOR, adjusted odds ratio; 95% CI, confidence interval.

while significant HBV protection was seen in HCWs who had adequate PPE and those using gloves and dust coat consistently. A study in Southern Ethiopia found HBV lifetime exposure was higher in MWH older than 40 years [5] while population based Azar cohort study found that all age groups were exposed to HBV. In in Eastern Ethiopia, there was higher prevalence of HBV infection in trainees [38, 39]. There is need to have remedial measures that is aimed at reducing this high lifetime exposure rates by capacity building HCWs, availing proper

**Table 5. Factors associated with lifetime exposure to HBV infection among health care workers in Kisumu County, 2020.**

| Characteristics | HCW (%) | N (%) | OR (95%CI) | p-value | aOR (95% CI) | p-value |
|---|---|---|---|---|---|---|
| **Sex** | | | | | | |
| Female | 92 (47.9) | 40 (43.5) | 1 | | | |
| Male | 100 (52.1) | 46 (46.0) | 1.11 (0.63–1.96) | 0.726 | | |
| **Age** | | | | | | |
| 20–29 | 65 (33.9) | 25 (38.5) | 1 | | 1 | |
| 30–39 | 83 (43.2) | 35 (42.2) | 1.17 (0.60–2.26) | 0.649 | 1.18 (0.37–3.77) | 0.784 |
| 40–49 | 23 (12.0) | 14 (60.9) | 2.49 (0.94–6.60) | 0.067 | 1.77 (0.28–11.37) | 0.547 |
| 50–59 | 18 (9.4) | 10 (55.6) | 2.00 (0.70–5.75) | 0.198 | 3.41 (0.33–34.81) | 0.301 |
| > = 60 | 3 (1.6) | 2 (66.7) | 3.2 (0.28–37.15) | 0.352 | 0.27 (0.01–7.98) | 0.452 |
| **Marital status** | | | | | | |
| Single | 37 (19.3) | 15 (40.5) | 1 | | | |
| Married | 151 (78.7) | 69 (45.7) | 1.23 (0.59–2.56) | 0.572 | | |
| Widowed/Separated/Divorced | 4 (2.1) | 2 (50.0) | 1.47 (0.19–11.59) | 0.716 | | |
| **Year of service** | | | | | | |
| Less than 1 year | 8 (4.2) | 6 (75.0) | 1 | | 1 | |
| 1–5 years | 85 (44.3) | 31 (36.5) | 0.19 (0.04–1.01) | 0.051 | 0.13 (0.01–2.79) | 0.193 |
| 6–10 years | 51 (26.6) | 26 (51.0) | 0.35 (0.06–1.88) | 0.22 | 0.57 (0.02–15.34) | 0.741 |
| More than 10 years | 48 (25.0) | 23 (47.9) | 0.31 (0.06–1.67) | 0.172 | 0.31 (0.01–9.82) | 0.509 |
| **Carder** | | | | | | |
| Doctors | 8 (4.2) | 3 (37.5) | 1 | | 1 | |
| Clinical Officer | 29 (15.1) | 14 (48.3) | 1.56 (0.31–7.75) | 0.59 | 0.72 (0.05–10.26) | 0.811 |
| Nursing officer | 39 (20.3) | 15 (38.5) | 1.04 (0.22–5.01) | 0.959 | 0.49 (0.04–6.59) | 0.589 |
| Laboratory technologist | 55 (28.7) | 18 (32.7) | 0.81 (0.17–3.78) | 0.789 | 0.47 (0.05–4.87) | 0.527 |
| Mortuary attendance | 7 (3.7) | 4 (57.1) | 2.22 (0.28–17.63) | 0.45 | 0.39 (0.02–8.56) | 0.578 |
| HTS counsellor | 17 (9.9) | 8 (47.1) | 1.48 (0.27–8.27) | 0.654 | 0.73 (0.05–9.71) | 0.811 |
| Waste handlers | 37(19.3) | 24(64.9) | 3.08(0.63–14.98) | 0.164 | 2.00(0.17–22.84) | 0.578 |
| **HBV Vaccine uptake** | | | | | | |
| Not vaccinated | 78 (40.6) | 52 (66.7) | 1 | | 1 | |
| 1 dose | 30 (15.6) | 25 (83.3) | 2.5 (0.86–7.28) | 0.093 | 6.25 (1.29–30.30) | 0.023 |
| ≥2 doses | 84 (43.8) | 9 (10.7) | 0.06 (0.03–0.14) | <0.0001 | 0.03 (0.01–0.10) | <0.0001 |
| **History of exposure** | | | | | | |
| Needle Stick Injury | 91 (47.4) | 37 (40.7) | 1 | | 1 | |
| Contact Exposure | 11 (5.7) | 2 (18.2) | 0.32 (0.07–1.59) | 0.165 | 0.10 (0.01–1.40) | 0.087 |
| No Known Exposure | 90 (46.9) | 47(52.2) | 1.59 (0.89–2.87) | 0.12 | 3.55 (1.34–9.43) | 0.011 |
| **Blood transfusion** | | | | | | |
| No | 189 (98.4) | 85 (45.0) | 1 | | | |
| Yes | 3 (1.6) | 1 (33.3) | 0.61 (0.05–6.86) | 0.69 | | |
| **Knowledge on HBsAg Transmission** | | | | | | |
| No | 83 (43.2) | 43 (51.8) | 1 | | 1 | |
| Yes | 109 (56.8) | 43 (39.5) | 0.61 (0.34–1.08) | 0.089 | 17.97 (2.10–153.39) | 0.008 |
| **Knowledge on HBsAg Prevention and control** | | | | | | |
| No | 89 (46.4) | 46 (51.7) | 1 | | 1 | |
| Yes | 103 (53.7) | 40 (38.8) | 0.59 (0.33–1.05) | 0.075 | 0.34 (0.06–2.12) | 0.25 |
| **Training on PPE full module** | | | | | | |
| No | 141 (26.6) | 65 (46.1) | 1 | | | |
| Yes | 51 (73.4) | 21 (41.2) | 0.82 (0.43–1.57) | 0.545 | | |
| **Training on infectious agent on waste full module** | | | | | | |

(*Continued*)

**Table 5.** (Continued)

| Characteristics | | HCW (%) | N (%) | OR (95%CI) | p-value | aOR (95% CI) | p-value |
|---|---|---|---|---|---|---|---|
| | No | 130 (67.7) | 64 (49.2) | 1 | | 1 | |
| | Yes | 62 (32.3) | 22 (35.5) | 0.57 (0.30–1.06) | 0.075 | 0.65 (0.10–4.21) | 0.652 |
| **Training on waste management full module** | | | | | | | |
| | No | 134 (69.8) | 65 (48.5) | 1 | | 1 | |
| | Yes | 58 (30.2) | 21 (36.21) | 0.60 (0.32–1.14) | 0.117 | 2.37 (0.32–17.28) | 0.395 |
| **Knowledge on waste disposal** | | | | | | | |
| | No | 90 (46.9) | 47 (52.2) | 1 | | 1 | |
| | Yes | 102 (53.1) | 39 (38.2) | 0.57 (0.32–1.01) | 0.053 | 0.41 (0.11–1.47) | 0.17 |
| **Availability of PPEs** | | | | | | | |
| | No | 18 (9.4) | 7 (38.9) | 1 | | | |
| | Yes | 174 (90.6) | 79 (45.4) | 1.31 (0.48–3.53) | 0.598 | | |
| **Adequacy of PPEs** | | | | | | | |
| | No | 149 (77.6) | 74 (49.7) | 1 | | 1 | |
| | Yes | 43 (22.4) | 12 (27.9) | 0.39 (0.19–0.82) | 0.013 | 0.49 (0.16–1.56) | 0.228 |
| **Daily use of apron/Dust coat** | | | | | | | |
| | No | 67 (34.9) | 37 (55.2) | 1 | | 1 | |
| | Yes | 125 (65.1) | 49 (39.2) | 0.52 (0.29–0.95) | 0.034 | 0.73 (0.26–2.03) | 0.549 |
| **Daily use of gumboots (Waste handlers only)** | | | | | | | |
| | No | 32 (86.5) | 22 (68.8) | 1 | | | |
| | Yes | 5 (13.5) | 2 (8.3) | 0.82 (0.13–5.01) | 0.827 | | |
| **Daily use of gloves** | | | | | | | |
| | No | 106 (55.2) | 56 (52.8) | 1 | | 1 | |
| | Yes | 86 (44.8) | 30 (34.9) | 0.48 (0.27–0.86) | 0.013 | 0.40 (0.16–1.01) | 0.052 |
| **Daily use of mask when on duty** | | | | | | | |
| | No | 84 (43.8) | 40 (47.6) | 1 | | | |
| | Yes | 108 (56.3) | 46 (42.6) | 0.82 (0.56–1.45) | 0.487 | | |
| **Access to hand hygiene** | | | | | | | |
| | No | 7 (3.7) | 3 (42.9) | 1 | | | |
| | Yes | 185 (96.4) | 83 (44.9) | 1.08 (0.24–4.98) | 0.917 | | |
| **Availability of waste management materials (All 3 waste bins)** | | | | | | | |
| | No | 8 (4.2) | 4 (50.0) | 1 | | | |
| | Yes | 184 (95.8) | 82 (44.6) | 0.80 (0.20–3.31) | 0.763 | | |
| **Proper waste segregation** | | | | | | | |
| | No | 135 (70.3) | 66 (48.9) | 1 | | 1 | |
| | Yes | 57 (29.7) | 20 (35.1) | 0.57 (0.30–1.07) | 0.081 | 0.94 (0.28–3.19) | 0.991 |
| **Waste disposal Method (Incineration)** | | | | | | | |
| | No | 63 (32.8) | 28 (44.4) | 1 | | | |
| | Yes | 129 (67.2) | 58 (45.0) | 1.02 (0.56–1.87) | 0.946 | | |

Values are presented as number (%); OR odds ratio; aOR, adjusted odds ratio; CI, confidence interval.

infrastructure for infection prevention and control, strengthening HBV vaccination and proper surveillance for HAI in all healthcare settings. Policy should be revised to enforce mandatory HAI pre-employment screening and vaccination for personnel working in healthcare settings.

All the study participants got their results, the 36 participants positive to HBV infections were referred for clinical review and further management, the 49 who had resolved infections and 72 who were immunized were counseled and reassured while the 34 who were susceptible

got their vaccination through KEPI. There was one participant who had isolated core antibody, this could have resulted from: (i) false anti-HBc positive test, (ii) past HBV infection, (iii) occult HBV infection [40]. There was need to know if the participant with isolated core antibody had ever become immunosuppressed or had chemotherapy, this participant was referred to clinicians for further clinical management (Table 2). Additional studies may be required to determine the causes, effects, and prevalence of isolated anti-HBc among healthcare workers.

Limitations for this study were that this was done during COVID 19 outbreak period when there was a lot of fucus on infection prevention to mitigate COVID 19 infections, in as much as this may have influenced the findings on the compliance with standard precautions, it strengthened the adherence to standard infection prevention measures. Also, during this COVID 19 period the government issued recommendations that elderly populations and those with comorbidities should work from home so we may have missed some eligible health care workers at the facilities. The study was conducted in nine highest volume government hospitals, these hospitals have the highest workload, produce the largest volume wastes, and have highest number of health care workers. However, results from this purposive sample of facilities cannot be generalized across all health facilities nationally and therefore, a national survey covering both public and private facilities would provide a more precise estimate of HBV infections among HCW in Kenya. We only assessed HBV exposures that are related to health care settings therefore the generalizability is limited to the studied health facility related exposures. Health care workers are not done for thorough pre-employment medical examination so it's difficult to point if the infections observed in this study occurred before or after employment.

## Conclusion

The prevalence of HBV infections among HCW in these nine facilities was about 6 and 5-fold higher than general population and adolescent blood donners respectively, this high prevalence needs multi stakeholder approach to address. There was suboptimal training on waste management, infectious agent on waste and PPE coupled with PPE inadequacy that could lead to high needle stick injury poor adherence to universal/standard precautions.

There is need to ensure that adequate PPEs is available for HCW usage, trainings done to health care workers on infection prevention and control.

The study low HBV vaccine uptake and low dose completion rate, high infection rate was observed in HCWs who have neither been vaccinated nor completed the dosage significant relationship was also observed between immunization status and positivity for HBV. The high lifetime exposure may be due to high exposure to infections and low vaccination rates for personnel working in healthcare settings, policy should be revised to make it mandatory for pre-employment HBV vaccination for most at risk populations like HCWs.

No significant association was observed between HBV exposure and factors such as a history of exposure, blood transfusion, use of PPE, Knowledge on HBsAg (transmission, pathogenicity, treatment, prevention & control), training on (PPE, infectious agent, waste management). None of the sociodemographic characteristics plus other factors such as carder and departments were significantly associated with HBV exposure status for HCWs, there is need for proper surveillance for HAI in all healthcare settings.

## Recommendation

There should be continuous training of HCW on universal precaution of infection prevention measures. There is need to increase HBV vaccine coverage and improve HBV surveillance among HCW.

## Acknowledgments

We acknowledge the county Government of Kisumu, each hospital's medical superintendent and management for allowing us to do this study. We are also grateful to Kenya medical research institute, Center for Global HIV research, Human immunodeficiency Virus Research Laboratory (KEMRI, CGHR, HIV-R Laboratory) for performing the laboratory tests for the study.

We sincerely thank the data collectors and laboratory Scientists of each hospital for their assistance in data and sample collection.

We thank and appreciate the study subjects who volunteered to participate in this study. Lastly, we wish to appreciate and thank of Jaramogi Oginga Odinga University of Science and Technology for their mentor and support in designing and conducting the study.

## Author Contributions

**Conceptualization:** Frankline Otieno Mboya, Ibrahim I. Daud, Daniel Onguru.

**Data curation:** Frankline Otieno Mboya.

**Formal analysis:** Frankline Otieno Mboya, Raphael Ondondo, Daniel Onguru.

**Funding acquisition:** Frankline Otieno Mboya.

**Investigation:** Ibrahim I. Daud.

**Methodology:** Frankline Otieno Mboya, Ibrahim I. Daud, Raphael Ondondo, Daniel Onguru.

**Project administration:** Frankline Otieno Mboya, Ibrahim I. Daud.

**Resources:** Frankline Otieno Mboya, Daniel Onguru.

**Software:** Raphael Ondondo, Daniel Onguru.

**Supervision:** Frankline Otieno Mboya, Raphael Ondondo, Daniel Onguru.

**Validation:** Frankline Otieno Mboya, Raphael Ondondo, Daniel Onguru.

**Visualization:** Frankline Otieno Mboya, Ibrahim I. Daud, Daniel Onguru.

**Writing – original draft:** Frankline Otieno Mboya, Raphael Ondondo, Daniel Onguru.

**Writing – review & editing:** Frankline Otieno Mboya, Ibrahim I. Daud, Raphael Ondondo, Daniel Onguru.

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
