## [Decision Letter · Decision Letter 0]

27 Mar 2023

PGPH-D-22-02115

Hepatitis B virus infection status and associated factors among health care workers in selected hospitals in Kisumu County, Kenya: a cross-sectional study.

Dear Dr. Mboya,

Thank you for submitting your manuscript to PLOS Global Public Health. After careful consideration, we feel that it has merit but does not fully meet PLOS Global Public Health’s publication criteria as it currently stands. Therefore, we invite you to submit a revised version of the manuscript that addresses the points raised during the review process. Please pay careful attention to the methods section.

We look forward to receiving your revised manuscript.

Kind regards,

Esmita Charani, MPharm, MSc

Academic Editor

Journal Requirements:

Additional Editor Comments (if provided):

Dear authors, having read the manuscript and considering the reviewer comments, I agree with the reviewers that this manuscript whilst of interest to the journal needs to be undergo a major revision, particularly to the methods section. We invite you to address the comments of the reviewers.

Reviewers' comments:

Reviewer's Responses to Questions

**Comments to the Author**

1. Does this manuscript meet PLOS Global Public Health’s publication criteria? Is the manuscript technically sound, and do the data support the conclusions? The manuscript must describe methodologically and ethically rigorous research with conclusions that are appropriately drawn based on the data presented.

Reviewer #1: Partly

Reviewer #2: Partly

2. Has the statistical analysis been performed appropriately and rigorously?

Reviewer #1: Yes

Reviewer #2: N/A

3. Have the authors made all data underlying the findings in their manuscript fully available (please refer to the Data Availability Statement at the start of the manuscript PDF file)?

Reviewer #1: Yes

Reviewer #2: Yes

4. Is the manuscript presented in an intelligible fashion and written in standard English?

Reviewer #1: Yes

Reviewer #2: No

5. Review Comments to the Author

Reviewer #1: This is a relevant study in a data-scarce and critical area for the African region. The study evaluates serological markers of HBV and related factors in health workers in Kisumu County -Kenya. Below are some comments and suggestions:

Introduction

1.I suggest including the vaccination strategy for HBV in Kenya (general population and healthcare workers).

Methods

2.Can you Detail the method used to choose the 192 HCWs? How many illegible health professionals were in each Health Unit and how were the 192 chosen??

3.The study was conducted from May 2020 to April 2021. What was the recruitment period?

4.Where were the sample tests performed? How was the quality of results ensured when testing for HBV serological markers?

Results

5.I suggest organizing table 2 into HCW infected (HBsAg positive), susceptible (negative for all markers), with immunity due to vaccination (Anti-HBs only), and immunity due to past infection (Anti-HBc and Anti-HBs). This change will affect the presentation of the results in the abstract, "prevalence of HBV Biomarkers" and discussion.

6.How many HBsAg positive participants had two and three doses of vaccine? Separating into two categories (with at least one dose of vaccination/unvaccinated) can help reduce the amplitude of the confidence interval.

Discussion

7.In the discussion of the anti-HBs results, I additionally suggest a separate discussion of immunity by vaccination.

8.It is interesting that a recently published study on HBV prevalence in HCWs in Mozambique also found high numbers of vaccinated laboratory technologists with the same justification. I suggest adding it to the discussion (line 261).

9.Correct the sentence in lines 273-274.

10.Correct the sentence in line 297 “facility related”

Conclusion

11.I suggest that the conclusion be summarized, with a focus on the significance and implication of the results found.

Reviewer #2: This is an interesting manuscript to review. I have reviewed the up to method section and found the method section needs rigorous improvement. After addressing the method section comments. There is a lot of space to improve the manuscript. I am happy to review the entire manuscript.

6. PLOS authors have the option to publish the peer review history of their article (what does this mean?). If published, this will include your full peer review and any attached files.

**Do you want your identity to be public for this peer review?** For information about this choice, including consent withdrawal, please see our Privacy Policy.

Reviewer #1: No

Reviewer #2: **Yes: **Md Golam Dostogir Harun

---

## [Decision Letter · Decision Letter 1]

10 Jul 2023

PGPH-D-22-02115R1

Hepatitis B virus infection status and associated factors among health care workers in selected hospitals in Kisumu County, Kenya: a cross-sectional study.

Dear Dr. Mboya,

Thank you for submitting your manuscript to PLOS Global Public Health. After careful consideration, we feel that it has merit but does not fully meet PLOS Global Public Health’s publication criteria as it currently stands. Therefore, we invite you to submit a revised version of the manuscript that addresses the points raised during the review process.

We look forward to receiving your revised manuscript.

Kind regards,

Max Carlos Ramírez-Soto, BSc, MPH, FRSPH, MACE

Academic Editor

Journal Requirements:

Additional Editor Comments (if provided):

Authors should review the reviewer's comments.

Reviewers' comments:

Reviewer's Responses to Questions

**Comments to the Author**

1. If the authors have adequately addressed your comments raised in a previous round of review and you feel that this manuscript is now acceptable for publication, you may indicate that here to bypass the “Comments to the Author” section, enter your conflict of interest statement in the “Confidential to Editor” section, and submit your "Accept" recommendation.

Reviewer #1: All comments have been addressed

Reviewer #2: (No Response)

2. Does this manuscript meet PLOS Global Public Health’s publication criteria? Is the manuscript technically sound, and do the data support the conclusions? The manuscript must describe methodologically and ethically rigorous research with conclusions that are appropriately drawn based on the data presented.

Reviewer #1: Yes

Reviewer #2: No

3. Has the statistical analysis been performed appropriately and rigorously?

Reviewer #1: Yes

Reviewer #2: N/A

4. Have the authors made all data underlying the findings in their manuscript fully available (please refer to the Data Availability Statement at the start of the manuscript PDF file)?

Reviewer #1: Yes

Reviewer #2: Yes

5. Is the manuscript presented in an intelligible fashion and written in standard English?

Reviewer #1: Yes

Reviewer #2: Yes

6. Review Comments to the Author

Reviewer #1: (No Response)

Reviewer #2: This is good paper to review abut the paper has significant space to improve.I am happy to review the updated manuscript if the initial comments have been addressed.

7. PLOS authors have the option to publish the peer review history of their article (what does this mean?). If published, this will include your full peer review and any attached files.

**Do you want your identity to be public for this peer review?** For information about this choice, including consent withdrawal, please see our Privacy Policy.

Reviewer #1: No

Reviewer #2: No

---

## [Editor Report · Decision Letter 2]

25 Aug 2023

Hepatitis B virus infection status and associated factors among health care workers in selected hospitals in Kisumu County, Kenya: a cross-sectional study.

PGPH-D-22-02115R2

Dear MR Mboya,

We are pleased to inform you that your manuscript 'Hepatitis B virus infection status and associated factors among health care workers in selected hospitals in Kisumu County, Kenya: a cross-sectional study.' has been provisionally accepted for publication in PLOS Global Public Health.

Best regards,

Max Carlos Ramírez-Soto, BSc, MPH, FRSPH, MACE

Academic Editor